# Creating the Internet of Augmented Things: An Open-Source Framework to Make IoT Devices and Augmented and Mixed Reality Systems Talk to Each Other [note 1]

**DOI:** 10.3390/s20113328

**Published:** 2020-06-11

**Authors:** Óscar Blanco-Novoa, Paula Fraga-Lamas, Miguel A. Vilar-Montesinos, Tiago M. Fernández-Caramés

**Affiliations:** 1Department of Computer Engineering, Faculty of Computer Science, Universidade da Coruña, 15071 A Coruña, Spain; o.blanco@udc.es; 2Centro de Investigación CITIC, Universidade da Coruña, 15071 A Coruña, Spain; 3Navantia S. A., Astillero de Ferrol, 15403 Ferrol, Spain; mvilar@navantia.es

**Keywords:** augmented reality, mixed reality, IoT, Internet of Things, open-source framework, Microsoft HoloLens, MQTT, sensors, actuators

## Abstract

Augmented Reality (AR) and Mixed Reality (MR) devices have evolved significantly in the last years, providing immersive AR/MR experiences that allow users to interact with virtual elements placed on the real-world. However, to make AR/MR devices reach their full potential, it is necessary to go further and let them collaborate with the physical elements around them, including the objects that belong to the Internet of Things (IoT). Unfortunately, AR/MR and IoT devices usually make use of heterogeneous technologies that complicate their intercommunication. Moreover, the implementation of the intercommunication mechanisms requires involving specialized developers with have experience on the necessary technologies. To tackle such problems, this article proposes the use of a framework that makes it easy to integrate AR/MR and IoT devices, allowing them to communicate dynamically and in real time. The presented AR/MR-IoT framework makes use of standard and open-source protocols and tools like MQTT, HTTPS or Node-RED. After detailing the inner workings of the framework, it is illustrated its potential through a practical use case: a smart power socket that can be monitored and controlled through Microsoft HoloLens AR/MR glasses. The performance of such a practical use case is evaluated and it is demonstrated that the proposed framework, under normal operation conditions, enables to respond in less than 100 ms to interaction and data update requests.

## 1. Introduction

The Internet of Things (IoT) paradigm has already been considered for multiple applications in fields like smart appliances [1,2], precision agriculture [3], smart healthcare [4,5] or smart buildings and cities [6]. In fact, some reports point at a huge growth on IoT deployments, with 75 billion devices in operation by 2025 [7]. Many IoT systems make use of web-based or smartphone apps to monitor and control them, which are adequate for most situations, but the latest advances of Augmented Reality (AR) and Mixed Reality (MR) can bring the interaction with such systems to a new level.

The first pioneering AR/MR developments were carried out in the 1960s [8,9], but they were not practical for being used massively. It was not until the 1990s, after Boeing documented the first industrial AR applications [10], when the field regained interest by industry and academia [11]. However, the big push for AR/MR came from the German government in the late 1990s, when it funded the ARVIKA project, which involved some of the largest industrial manufacturers (e.g., Airbus, EADS, BMW, Audi, VW, Daimler, Ford) in the development of mobile AR/MR systems [12,13].

Luckily, during the last years AR/MR devices have improved significantly thanks to advances on electronics, computing technologies and wireless communications. Such advances have also led to decrease the AR/MR commercialization price, which has sparked the interest of new consumers and industries that are transforming their processes through the Industrial IoT (IIoT) or Industry 4.0 paradigms [14].

One of the most relevant challenges that AR/MR developers currently face when interacting with IoT devices is the technology heterogeneity, which makes it difficult to implement even simple interactions. Thus, most AR/MR frameworks use technologies that differ remarkably to those used for the development of IoT device software. Moreover, the skills usually required by AR/MR and IoT experts are also different, what complicates the communication between both fields.

In order to allow AR/MR applications to interact with the surrounding IoT devices, it is necessary to develop communication mechanisms that enable exchanging data in the same ’language’, so that the involved devices understand each other. Unfortunately, the development of such mechanisms is not straightforward due to the previously mentioned heterogeneity issues, the computational hardware limitations (especially in the case of resource-constrained IoT devices) and the development restrictions imposed by AR/MR frameworks.

To tackle the aforementioned problems, this article includes the following contributions. First, it is faced the challenge of interconnecting very different types of systems by proposing an AR/MR-IoT framework that allows for integrating AR/MR platforms and IoT devices easily through the use of open-source standard communication protocols. Such a framework enables implementing ubiquitous and scalable applications in a flexible way and eases their configuration. In addition, the proposed framework is designed to allow for implementing on top of it complex functionality in a simple way. Such a development is eased by providing the framework software through GitHub together with implementation examples [15]. Moreover, this article follows a hands-on approach in order to lead practitioners and developers in the development of future AR/MR-IoT frameworks. Finally, a number of experiments are presented in order to evaluate the performance of the framework in terms of Quality of Experience (QoE). The results show that the framework is really fast, being able to perform interaction and data updates in less than 100 ms under normal operation conditions.

The rest of this article is organized as follows. Section 2 reviews the most relevant related works. Section 3 describes the design and implementation of the proposed AR/MR-IoT framework. Section 4 details how the framework can be used for implementing an energy monitoring and control application based on a smart power socket and Microsoft HoloLens smart glasses. Finally, Section 5 is dedicated to the evaluation of the response latency of the framework, while Section 6 is devoted to the conclusions.

## 2. Related Work

In the last years, novel smart environments have been conceived under what has been denominated as Extended Reality (XR). In such spaces, digital, physical, and social layers are strongly intertwined enabled by technologies such as AR/MR, Virtual Reality (VR) and immersive displays [16]. Such technologies offer more natural ways to exploit human perception and interaction [17], and provide enhanced data use [18]. Thus, they represent an opportunity to rethink the way in which people collaborate and interact nowadays. Examples of such collaborations are recent novel applications for telemedicine [19], shared mission planning scenarios [20] or for enhancing astronaut procedure execution [21].

The IoT is a field where protocol, standard and hardware compatibility problems are common due to the diversity of manufacturers and developers [22]. The mentioned problems also affect AR/MR interfaces, but only a few academic articles have addressed them. An example is the work of Croatti et al., who detail the main components of an infrastructure for supporting what they called the Web of Augmented Things (WoAT) [23]. Another example is presented in [24], where a comprehensive review on the main challenges for enhancing AR/MR-IoT integration is provided. In such a review the authors emphasize the issues that arise when managing and visualizing distributed object-centric data; when accessing, controlling and interacting with IoT objects; and when carrying out interoperable content exchanges. Moreover, the same authors presented in [25] a proof-of-concept demonstrator to enhance shopping experiences. Another demonstrator is presented in [26], but in such a case for monitoring stress analyses of metal shelving. It is also worth mentioning other implementations like the one detailed in [27], where the authors studied the problems related to the automatic discovery of IoT sensor devices and proposed to solve them by using relational localization mechanisms.

Regarding AR/MR-IoT frameworks, four relevant works can be cited. The first one is described in [28], where the authors present an AR framework whose tracking algorithm depends on the context, which is detected automatically by the IoT infrastructure. The second relevant work is detailed in [29]: the researchers propose to integrate Microsoft HoloLens smart glasses [30] with an IoT platform through a RESTful API. The third work focuses on integrating IoT devices and virtual elements with the aim of creating 4D experiences under the concept of Virtual Environment of Things (VEoT) [31]. Such real-time experiences enable users to interact with IoT networks through avatars and holograms that represent digital twins of sensing devices. Such a VEoT concept is demonstrated at the XReality lab of Texas State University [32], which provides simplified examples of object recognition, IoT-to-hologram interaction and network topology visualization.

Additionally, it should be noted that other works focused on facing the specific challenges that pose client-server infrastructures, IoT resource-constrained devices, mobility, interaction methods, data management or ad-hoc 3D streaming. One of such works is described in [33], where the authors propose a hybrid framework that facilitates 3D texture streaming, which provides a high quality service with limited power consumption.

### 2.1. Quality of User Experience

User eXperience (UX) is a key factor to consider when designing an AR/MR-IoT framework since UX will determine to a great extent the likelihood of user adoption. Quality of Experience (QoE) evaluation has been an active research topic for the past years in the AR/MR field. Academic works range from the QoE evaluation of basic AR systems [34,35] to more sophisticated approaches that evaluate the use of HMDs [36,37] with sophisticated physiological metrics [37,38]. For instance, in [34] the authors compare the UX between an AR manual workstation and a video for assembly assistance. Their aim was to determine the probability of user adoption of an AR interface for such a use case. The obtained results show a better task performance using an AR interface, a reduction in both time to completion and in the number of errors. An animated AR system was also compared with a paper-based manual system as a guidance tool for an assembly task in [35]. Such a work conducted formal experiments with 50 participants, whose cognitive workload and learning curve were measured when using the system. The results of the experiments showed that the AR solution achieved less errors, shorter time to completion and lower total workload. Furthermore, the learning curve of the trainees improved significantly.

One of the most complete evaluations of an HMD was performed in [36]. Such a work evaluated the QoE of an AR system for solving a 3 × 3 Rubik’s Cube (an NP-complete problem with 43 quintillion possible states) and compared it with when using paper-based instructions. The experimental methodology included the analysis of both implicit and explicit QoE metrics for the use case of task assistance. Such a methodology involves six phases: sampling, screening, baseline, training, practice and testing. The task performance was objectively measured using task success rate and time to completion. Furthermore, in order to infer emotional state during task realization, the implicit metrics of electrodermal activity (EDA), heart rate, skin temperature and facial action units were collected.

Finally, with respect to explicit metrics, it was used a Likert scale questionnaire to subjectively report QoE under six variables: utility, usability, interaction, aesthetics, efficiency and acceptability. Emotional state was reported by participants using a Self-Assessment Manikin (SAM) questionnaire to reflect their emotions upon task realization. The SAM questionnaire involves three scales, one for each dimension of affect (i.e., arousal, valence and dominance). The results highlight the potential of AR with respect to efficiency and productivity: AR yielded higher success rates, significantly shorter time to completion, more positive emotions and less stressed states than paper-based instructions. However, users remark some issues with respect to aesthetics, therefore acknowledging the importance of a human-centered AR design.

Another work worth mentioning studied the user experience of a visual analytics system for the drifting data of bees [37]. In such a work, Microsoft HoloLens was compared with a Windows desktop interface through a correlation analysis of the UX ratings collected from 14 test subjects through a 30-min questionnaire using a 7-point Likert scale. The experiment included training, task solving and user feedback. Such a UX feedback evaluated criteria that included how intuitive, easy to use, comfortable, natural and efficient the interfaces are and obtained an overall rate of their interface preference. The study concluded that all the criteria were strongly correlated with the interface preference rating. Efficient was the most strongly correlated with interface preference. This fact suggests that end users, in case of a specific task to solve, they give importance to the efficiency to solve it. Comfortable was the least strongly correlated with interface predilection. Furthermore, while intuitiveness was highly correlated with the use of AR interfaces, the natural metric was highly correlated with the desktop interface. The authors suggest that participants with previous experience with HoloLens found the AR interface intuitive. As a result, such participants may have had a better experience as opposed to first-time users. According to our previous experience with Industry 4.0 operators [39,40], this assumption is highly probable.

Taking into account the results of the previous QoE works, it can be concluded that efficiency is one of the most important criteria regarding AR/MR user experience. Since other criteria for QoE assessment like effectiveness or satisfaction are strongly dependent on the AR/MR-IoT application, this work will only focus on efficiency, aiming to minimize task completion time by optimizing user interactions. Such an efficiency, when considering an AR/MR-IoT framework, is strongly correlated with throughput and latency values. Therefore, Section 5 will be devoted to the evaluation of the performance of the proposed framework regarding such parameters.

After reviewing the state of the art it can be concluded that there are only a few previous works with similar aims respect to the proposed AR/MR-IoT framework. However, in contrast to the work presented in this article, the vast majority of the previous works presented are very early developments with a relevant number of open issues that require further research.

## 3. Design and Implementation of the System

The main problem that the proposed framework aims to solve is to reduce the existing difficulties to interconnect heterogeneous systems such as those found in IoT and AR/MR devices. For such a purpose, this framework defines different mechanisms and methods to interconnect diverse technologies by using standardized protocols that can be easily integrated into both the target platforms and existing projects.

This section describes the challenges of a AR/MR-IoT system, the requirement analysis of the system as well as the design of each layer of the framework and how communications are handled by each component of the system.

### 3.1. Challenges of Implementing an AR/MR-IoT System

The process of implementing an AR/MR-IoT framework involves different challenges that are often overlooked and that should be considered during the design phase. As it was previously mentioned, one of main challenges is the heterogeneity of the technologies that need to be used in order to build a complete AR/MR-IoT system. This usually involves making low-level decisions about the architecture and software on diverse areas of knowledge.

Another important challenge is latency. In order to provide a proper user experience, it is necessary to keep response times low for the different communication protocols, component interactions and processing tasks. If a part of the system is related to delays in the information flow, a bottleneck will exist that will eventually affect the whole system, harming its usability.

Both AR/MR and IoT systems are usually complex, which makes difficult to integrate a new communication architecture into the existing software. This means that the technologies used by an AR/MR-IoT framework to implement the different interactions should be able to be integrated easily into existing projects. This can be achieved by providing well-documented open-source software.

Some of the works previously analyzed in Section 2 address some of the mentioned challenges, but it was not found in the literature any work that tackles all of them. Thus, Table 1 compares the features of the proposed framework with the ones of the three most similar previous works.

### 3.2. Requirements

As it was previously mentioned, one of the most important characteristics of an AR/MR application is the response time to the actions carried out by a user. If the response time is high, the user experience will not be good and the interaction would turn out to be less intuitive and pleasant. In addition, there are other requirements imposed by the type of devices used to implement the system as well as by the cost of implementation. Specifically, the main requirements for a AR/MR-IoT framework are the following:Latency. This is the metric that makes it possible to quantify how fast the system can react to a user actions. According to the International Telecommunication Union (ITU), the following latency times should be considered when assessing visual interaction performance [41]:Human reaction time (i.e., the latency between a human sensing a stimulus and responding with a muscular reaction), is roughly 1 s.Human auditory reaction time is about 100 ms.Human visual reaction time is in the range of 10 ms.An interface is said to have a fast reaction time if it is close to 1 ms.However, it is important to note that the previously indicated latencies are ideal approximations and in practice, when using AR/MR devices, they vary depending on the person, on the type of interaction and on the characteristics of the visualized content. For instance, the indicated human visual reaction time is aimed at providing a good experience when visualizing fast moving objects. In practice, most of the elements presented through an AR/MR interface do not move fast, so a good user experience can be provided despite the existence of latencies higher than 10 ms.Compatibility. The system must be able to interact with heterogeneous devices transparently. In addition, the data types required by an AR/MR interface are very different from those usually required by an IoT device, so an AR/MR-IoT framework must adapt the protocols to the needs of each device.Ease of development. An AR/MR-IoT framework implementation should be easy to replicate and flexible enough to make it easy to modify it to be adapted to new use cases. The upper layers of the framework should allow for such a flexibility. For instance, in the framework presented in this article, the use of a software like Node-RED makes it possible to change data flows graphically, by just changing nodes through a visual interface.Ease of integration. The framework should be easy to integrate within existent applications. As it was mentioned before, the characteristics of current AR/MR and IoT systems are very different, so the framework should be able to integrate every AR/MR and IoT seamlessly.

### 3.3. Communications Architecture

Figure 1 depicts the communications architecture of the AR/MR-IoT framework. As it can be observed, it consists of two main layers: the IoT Node Layer and the AR/MR-IoT Layer. The former is divided into two sublayers:The AR/MR Device Sublayer is composed by the diverse AR/MR devices (e.g., smart glasses, smartphones, tablets), which are able to exchange information with the AR/MR-IoT Layer through a wireless Access Point (AP). The pink arrows of Figure 1 represent the communications related to AR/MR devices.The IoT Device Sublayer is formed by the different IoT networks managed by the system. Such networks are composed essentially by sensors and actuators, which may be part of other systems like smart appliances, industrial machinery or home automation systems. The green arrows of Figure 1 indicate the main paths followed by IoT device communications. Note that the communications topology represented in Figure 1 is a mesh, but other topologies may be considered (e.g., a star).

Regarding the AR/MR-IoT Layer, it is responsible for interconnecting AR/MR and IoT devices. It may be implemented on a remote cloud on the Internet or on a local edge computing server (e.g., a fog computing node or a cloudlet [40,42]). Due to restrictions imposed by the compatibility requirement, the system has to be designed to be interoperable, as it has to be able to allow for communicating very heterogeneous devices. For such a purpose, it was designed to use standard protocols supported by a wide range of devices and applications.

It may be tempting to try to implement the entire system using IoT communication protocols like Message Queuing Telemetry Transport (MQTT), but, unfortunately, they are currently not appropriate for implementing AR/MR-IoT applications: such applications often require sending large amounts of information (e.g., related to the displayed 3D models or to real-time streams of data) that are difficult to handle through MQTT and similar IoT protocols, which have been designed for managing small payloads. This would not fulfill the ease-of-use requirement, since designing a protocol for managing large amounts of data through the mentioned IoT protocols is usually complicated and not very efficient.

Moreover, in some AR/MR development environments it is not easy to make use of protocols like MQTT due to the restrictions imposed by the available AR/MR frameworks and high-level APIs. As the ease-of-integration requirement indicates, the framework should be integrated easily, but the cost of implementing MQTT or similar protocols on top of proprietary APIs, if possible, would be higher than other alternatives already available for the target AR/MR framework. Nonetheless, according to the framework requirements, protocols like MQTT can be used only for communicating IoT devices, while other solutions need to be devised to communicate such devices with AR/MR platforms. Therefore, a mixed approach is required to fulfill all the framework requirements.

Considering the previous aspects, this article proposes a design that is essentially composed by the following software components:REST Application Programming Interface (API). It is in charge of managing the data exchanges related to the AR/MR devices. Many AR/MR frameworks offer a very high-level programming API that make it difficult to access low-level communications functionality (in some cases it is even impossible). A REST API ensures compatibility with most of the existing AR/MR frameworks in a simple way and it is flexible enough to be added to existing projects with little effort, thus fulfilling the ease-of-integration requirement.MQTT broker. It allows for implementing a publish/subscribe service that enables IoT devices to send and to receive data asynchronously. It is based on MQTT, which is an open standard protocol that is really lightweight, so it can be easily implemented even in resource-constrained IoT devices. This protocol is widely used in heterogeneous IoT networks and it is supported by many IoT platforms. It also provides other appealing features for IoT networks like low-latency, low-power and delay tolerance.AR/MR-IoT Bridge Service. This component is essential for the whole framework, since it is the one responsible for actually interconnecting the different AR/MR and IoT components. As the ease-of-development requirement states, it is essential for this component to be easily configurable so that it can be changed to fit each application. In practice, this component delivers the collected IoT data to the AR/MR devices that request them and sends commands from such AR/MR devices to IoT devices. Apart from protocol translation, it also performs caching operations, storing all recent data values shared by the IoT devices. Thus, response times are reduced since data are always available when a user application requests them, even when the device is in low-power mode, temporarily busy or unavailable. Furthermore, the exchanged data flows are processed within this component so that each device receives the information in the most appropriate format (thus fulfilling the framework compatibility requirement).

### 3.4. Component Model

Figure 2 shows the component model of the system, which illustrates the main inputs, outputs and processing components of the AR/MR framework, so that further research can be carried out based on it. Regarding the inputs, they include multiple data sources that generate information that is later fed into the processing components. Such data sources are:Surrounding IoT devices: they generate data on their sensors and internal state.AR/MR device sensors: they collect information from their embedded sensors, which allow for detecting where the user currently is, where he/she has previously been, what is he/she looking at or his/her posture.User profile: the framework can make use of relevant information on the user in order to later use it to determine the potential actions that he/she can carry out or the information he/she can access to. For example, in a smart home, it may be interesting to limit the actions that children can perform on certain appliances. Moreover, a user profile can be linked to past actions with the objective of using them as contextual information for future actions. For instance, the framework could infer the habits of a user from his/her past actions (i.e., from the sequence of his/her daily actions) and prepare the IoT devices with which the user interacts more frequently before the user requires their use (thus decreasing interaction latency and decreasing energy consumption).External inputs: certain external inputs, either from remote users or external services (e.g., Network Time Protocol (NTP) or weather forecast services) can influence when and how certain actions are performed on IoT or AR/MR devices. For example, the interactions of an AR/MR user on an IoT device may be different if they are performed during daytime or at night.System configuration parameters: the administrator of the system can establish specific rules and thresholds that determine how AR/MR user interactions are carried out on IoT devices.

With respect to the processing components shown in Figure 2, they gather information from the inputs and perform certain actions as an output. Specifically, Figure 2 includes the main processing elements previously illustrated in Figure 1: an MQTT broker, a REST API, a bridge service and auxiliary services. The proposed framework makes use of rules to process the data collected from the inputs, but the reader can easily observe that such inputs can fed other advanced processing components, like machine learning, deep learning or other artificial intelligence modules, which can fuse the input information and take specific decisions.

Finally, regarding the outputs illustrated in Figure 2, they are related to the interaction with surrounding IoT devices (i.e., they allow for acting on them or for collecting certain information from them) and with AR/MR devices (i.e., for displaying certain content or for storing certain data on the AR/MR device memory for their later use).

### 3.5. Support for Complex Functionality

The proposed framework was designed in such a way that it allows for implementing on top of it complex functionality in a simple way. An example of a complex functionality is related to the nature of many IoT devices: they require to implement an energy consumption policy to provide an adequate compromise between response time and low-power consumption. The proposed AR/MR-IoT framework allows, for instance, to make use of the location information of IoT and AR/MR devices to warn the former when an AR/MR user is close, so that IoT devices are ready in case the user decides to interact with them. Location can be determined by using the surrounding WiFi access points to which the different devices are connected to. The gathered location information can be sent to the framework Bridge Service, which stores it and takes the corresponding actions when a user moves from one area to another.

Figure 3 shows a sequence diagram that illustrates how the multiple involved entities would communicate among them. Regarding the involved IoT devices, they first indicate the access point to which they are connected to during their initialization. The AR/MR devices act in the same way when a user moves from one area to another. The Bridge Service is in charge of notifying (by using a specific MQTT topic) the devices that are in a certain area when a user is in its vicinity.

An IoT device can remain in a low-power consumption mode until it receives a notification that indicates that an AR/MR user is in the vicinity. Then, it changes its energy saving configuration and starts to make more frequent updates, so response times are significantly reduced. In addition, when a user moves away from an IoT device, such a device can be warned so as to return to its low-power consumption mode.

Figure 3 also illustrates the case when *User A* sends a command to *Device 1*, which is not in a nearby area, so its response time is larger than when interacting with closer IoT devices. In contrast, Figure 3 depicts the case when *User B* sends a command to *Device 3*, which is in the vicinity, so its response time is reduced significantly.

Moreover, it can be observed in Figure 3 that IoT devices try to upload to the Bridge Service the data they collect every time they wake up. Since the Bridge Service is in charge of caching the collected information, it can send it directly to the users when they ask for it, instead of waking up IoT devices, thus reducing response times and decreasing IoT device power consumption.

### 3.6. Implementation

The theoretical architecture detailed in Section 3.3 was implemented as illustrated in Figure 4. All the necessary software to replicate the system is available on GitHub [15]. The next subsections provide details on the different layers and sublayers depicted in Figure 4.

#### 3.6.1. IoT Device Sublayer

The practical implementation of this sublayer takes into account that IoT devices are usually constrained in terms of computational power and that they have to be efficient in terms of energy when they rely on batteries. As a consequence, the communications protocols used by this sublayer have to be lightweight in relation to the amount of necessary computational resources and required power consumption [43]. Moreover, such protocols should enable mechanisms that allow for responding fast to the different events related to the requests from IoT and remote AR/MR devices. Furthermore, the mentioned protocols should be standard in order to foster interoperability in IoT device ecosystems.

The previously mentioned necessary features justify the use of MQTT, which has been recently standardized [44], is lightweight, it is already supported by many IoT devices and is really easy to use: IoT devices simply subscribe to MQTT topics and wait for messages addressed to them. Similarly, if an IoT device wants to send certain collected sensor data or notifications, it just has to publish them on the corresponding MQTT topic.

#### 3.6.2. AR/MR Device Sublayer

Among the diverse AR/MR hardware devices currently available [45,46], the most popular are smartphones and tablets due their sufficient computational power and low cost. Although such devices may provide practical AR/MR applications, they do not offer an experience as immersive as smart glasses and Head-Mounted Displays (HMDs), whose price has been decreasing during the last years.

As of writing, Microsoft HoloLens smart glasses [30] provide the best trade-off between affordability (although they are not cheap: around $ 3500), usability and AR/MR experience. The latter is achieved thanks to the HoloLens acceleration hardware and the provided software platform, which makes use of an operating system adapted to AR/MR development. Such a software platform eases the work of HoloLens developers, but it must be noted that certain low-level developments (e.g., related to raw video processing or communication sockets) are not supported directly by HoloLens Software Development Kit (SDK), so their use requires additional programming effort. In contrast, other communication protocols like HTTP are supported by the SDK, so their use is straightforward. Similarly, other actions that are very useful for creating attractive interactions and immersive experiences are handled out of the box by the SDK.

The most relevant HoloLens SDK modules used by the implemented application are summarized in Figure 5). Such modules interact with the HoloLens hardware so as to provide a good AR/MR QoE. The following are their main tasks:Gaze Manager and Gaze Stabilizer modules: they track the user head orientation and where a user is looking at.Spatial Mapping module: it obtains and updates a 3D-map of the user surroundings.Gesture Manager module: it recognizes the hand gestures performed by the user. In Figure 5 the black arrow that departs from the Gesture Manager indicates the path that is followed when a user interacts with the smart glasses: the Button Handler detects the gesture and then calls the Service Module of the AR/MR-IoT framework. If the detected action involves interacting with an IoT device, then the corresponding HTTP request is sent to the AR/MR-IoT Layer (to Node-RED [47], whose inner workings are detailed later).

#### 3.6.3. AR/MR-IoT Layer

This layer runs on a cloud an MQTT server called Mosquitto [48], Node-RED and a REST API. Thus, the layer routes the messages exchanged between the IoT and the AR/MR devices as follows:When an AR/MR device wants to perform an action on an IoT device (e.g., to collect certain data or to actuate on it), it sends an HTTP request to Node-RED through the REST API.The HTTP request is processed by Node-RED and then it decides to which IoT device it should be forwarded to. Such a decision is usually conditioned by a session token that is embedded on the AR/MR device request.The request is forwarded to the IoT device through an MQTT message that is published on Mosquitto under a specific topic.Since the target IoT device will be subscribed to the topic where the MQTT message was published, it will consume it as soon as such an IoT device receives the new message notification from Mosquitto.To send the data, the IoT device also makes use of MQTT messages through Mosquitto. Such messages are stored by Node-RED and then forwarded to the AR/MR devices that request for updates.

As it can be observed from the previous description, the core of the implementation is Node-RED, which enables interconnecting the different elements of the system. Node-RED is a visual-programming tool based on Node.js [49] that allows for managing easily the data flows that connect the multiple components. Therefore, Node-RED acts as a translator of the different protocols embedded into the exchanged messages.

## 4. A Practical Application: AR/MR Based Energy Monitoring and Control

In order to validate the proposed design and to assess the level of compliance of the framework requirements, a practical application was analyzed and designed to provide a reference implementation. Such an application consists of a smart power outlet for monitoring energy consumption that can be controlled from a pair of Microsoft HoloLens smart glasses. Specifically, the proposed system provides AR/MR users with real-time data collected from a commercial smart power outlet. Such users can also control the switching on and off of the power outlet by interacting with a virtual dashboard through gestures.

It must be noted that the proposed practical implementation was devised to provide fast response times so as to deliver a good user experience when using the AR/MR application. In addition, it was designed to be easily replicated and to asses easily the compatibility of IoT devices with the proposed framework. The next subsections detail the main components of the system.

### 4.1. IoT Smart Socket

The used IoT smart power socket is a commercial device (a Sonoff POW module, from Itead Studio [50]) that has a WiFi-enabled microcontroller (ESP8266) and a relay. It also embeds a current sensor (HLW8012) and a Shunt resistor to measure voltage. Figure 6 shows an example of test setup, where a smart socket controls and monitors a lamp. In addition, in Figure 6 there is a QR code label (it is usually glued to the top of the smart socket case): it is used by the HoloLens app to obtain the smart socket id fast, so that the HoloLens glasses can address their requests to such a specific smart socket through the AR/MR-IoT framework.

The smart socket can monitor the current that flows through the socket and log the intensity, so it can determine the instantaneous power and the consumed energy. In addition, the smart socket allows for sending ON and OFF commands through a standard MQTT communications channel. Such a channel is also used to send back the collected energy consumption values.

The smart socket firmware is based on Tasmota [51], which is one of the most advanced open-source firmware for smart appliances. The firmware has native support for MQTT, so the integration with the AR/MR-IoT framework is straightforward and does not require any further modifications other than configuring the connection parameters to connect to the MQTT broker server. However, the firmware was modified to support other features that allow for programming operating intervals, so that the smart socket can be switched ON and OFF automatically during specific time instants (e.g., when the electricity cost is cheaper). This latter feature requires to obtain and store the daily energy prices and then implement an intelligent planning algorithm that is capable of using the energy price data to make the appropriate decisions.

### 4.2. Node-RED Configuration and REST API

Node-RED is the technology responsible for the bridge service operations. It translates the information between the different protocols used to communicate with both the IoT and the AR/MR device sublayers. It also performs caching operations to improve the latency of the system and modifies data types to make the system interoperable. It was configured as it is shown in the screenshot in Figure 7. At the top of such a Figure are the system initialization, configuration and cleaning nodes that set the sampling periods and configure the associated smart sockets. Below them are the nodes that handle data storage and caching, as well as the REST API endpoints.

The connections between the nodes indicate the existing data flows while the nodes represent inputs and outputs as well as data processing nodes. The proposed setup makes it easy to reconfigure the data flows to fit new particular use cases, requiring little effort.

The REST API is composed of different endpoints that allow applications to perform all the necessary actions. All of them are exposed through Node-RED HTTP service and are intended for reading and writing data from the applications. A brief description of each of them is given in Table 2.

For security reasons, all endpoints accept as input parameters the smart socket identifier and a cookie with an authentication token that identifies the user.

### 4.3. MQTT Topics

The MQTT message hierarchy was structured around topics divided into three levels. The first level was related to the root MQTT topics, which were defined depending on the type of message to be handled:**/cmnd:** topics with this root are requests sent from a client to an IoT device.**/stat:** topics with this root are always responses to cmnd-type messages.**/tele:** topics with this root correspond to messages sent by the devices periodically to update their status.

The second topic level of the message hierarchy corresponds to the socket identifier, which includes the user token and the unique hardware identifier of the socket with the format *token_hwId*.

The last level of the hierarchy indicates an action or is related to certain information:**/sensor:** It is used for obtaining the consumption and power data of the socket.**/state:** It is used for collecting the socket status information (e.g., ON/OFF, uptime).**/result:** It is used to notify the result of the requests to the socket.**/timerX:** It is used to exchange information and actions on the timers.**/power:** It obtains the power status of the socket (i.e., 1 (ON) or 0 (OFF)).**/timers:** It enables sending together the information of all the 16 timers.**/prices:** It is used to send the energy price per hour list.

Finally, it is worth pointing out that the messages exchanged through the REST API and MQTT are JSON files, which allow for standardizing the data types defined by the system.

### 4.4. AR/MR Application

Figure 8 shows the main virtual dashboard of the developed AR/MR application, which can be moved by the user throughout the real-world scenario. The dashboard includes two virtual buttons to switch on and off the socket, a graph that shows the historical power consumption values and two displays that indicate the instantaneous consumed power and the power factor.

Users can interact with the dashboard by clicking on the virtual buttons through hand gestures. Such gestures are first captured by the HoloLens SDK and then processed by the Service Module of the AR/MR-IoT framework, which is the responsible for sending the switch on/off commands to the smart power outlet and for collecting from it the power consumption data.

All the communications that come from the AR/MR application make use of HTTP, which is natively supported by the HoloLens framework. This makes it possible to implement all the communications using the standard APIs provided by Microsoft without any further modifications. This fact, together with the use of MQTT for IoT device communications, allows the framework to provide wide compatibility, which would not be possible without the intervention of the bridge layer to fuse both protocols.

### 4.5. Use Cases

In order to describe the inner workings of the developed AR/MR-IoT smart socket system, the next subsections detail with the help of UML sequence diagrams several examples of relevant use cases.

#### 4.5.1. Real-Time Switching On and Off

Figure 9 shows the message flow required to perform a switch-on request. Such a request is initially sent via HTTP to the Node-RED REST API. The request is then translated to an MQTT query by an intermediate service and then sent through the MQTT broker to the smart power socket, which is switched on. Then the smart socket confirms the execution of the remote command by sending an acknowledgment message to Mosquitto, which forwards it to Node-RED. The same process is carried out for switching off the smart socket, but just changing the ON parameter for an OFF.

#### 4.5.2. Daily Hourly Energy Cost

To obtain the information on the energy cost per hour, in the case of Spain, it is necessary to access the API of Red Eléctrica Española (REE) [52]. Unfortunately, such an API returns an unfiltered XML file with a lot of information, whose parsing may be really slow, especially when it is carried out by resource-constraint IoT devices. To avoid this issue and thus speed up the process, it is used an intermediate service (called Pricing Service) that is in charge of filtering the information and converting it to a JSON file so that the information is much less verbose. This whole process is illustrated in Figure 10.

#### 4.5.3. Instantaneous Energy Values

This use case illustrates the process that is performed when the IoT smart socket notifies its status, which includes the following fields:ON/OFF status.Instantaneous consumption.Instantaneous current.Power factor.Total energy consumed today.Total energy consumed yesterday.Total energy consumed since the socket was installed.The time since the last reset of the smart socket.

The sequence diagram in Figure 11 illustrates the process for collecting the status of a smart socket. It is important to emphasize that Node-RED is in charge of obtaining the data from the sensors periodically and of caching them with the objective of releasing the storage load from the IoT socket. Thus, the status data are always available in the cache of the bridge layer, even though the IoT device may be in low-power mode, not available or even not connected to the Internet during certain time intervals. This feature is very important in order to fulfill the required response time requirement and therefore to provide an appropriate user experience on real-time AR/MR applications.

## 5. Experiments

The framework requirements indicated in Section 3.2 like ’compatibility’, ’ease of development’ and ’ease of integration’ can be assessed qualitatively in relation to the development experience during the implementation of the proposed smart socket-based application. However, this Section will be focused on the latency, which can be evaluated quantitatively by determining how low are response times (i.e., how a good is the user experience associated to interactions). As it was concluded in Section 2.1, UX is key in AR/MR applications: if the virtual objects respond slowly to the user interactions, the perceived experience will not be good.

Due to the need for achieving low-response times and providing that latency is a quantitative requirement that can be evaluated empirically, three sets of tests were performed to measure the response latency of the proposed AR/MR-IoT framework. The first set was aimed at estimating how fast the framework manages interaction requests. The second set of tests quantified the speed of the framework when updating the collected IoT data. Finally, the third set of tests analyzed the main interaction latencies for the use case detailed in Section 4.

As it can predicted, as the number of users and/or devices in the system increases, the load on the framework server increases accordingly, which impacts QoE. However, it is important to note that the load of each individual device remains constant regardless of the number of connected devices. This fact makes it possible to test the system emulating a large number of real devices, which would be really expensive (a pair of Microsoft HoloLens currently costs $ 3500).

To validate the use of emulated devices, 30 time-spaced requests were made to the framework first from a real device (a pair of Microsoft HoloLens glasses) and then from an emulated device while the framework server was idle. The obtained results are shown in Figure 12. As it can be observed in such a Figure, the average response time is very similar for both cases (96.159 ms for the emulated device and 95.636 ms for the HoloLens glasses) and, from the server point of view, the requests are completely indistinguishable, requiring the same computational load.

After validating the use of emulated devices, the rest of test environment was deployed. The cloud server used a virtual machine that run on a DELL Power Edge R415. Such a virtual machine had 4 GB of RAM and a 2-core AMD Opteron processor at 3.1 GHz. In addition, for both sets of tests, up to 500 devices were emulated in a desktop computer (i.e., a script executed on a desktop computer performed exactly the same requests as 500 real smart power sockets) with the objective of estimating the performance limits of the framework without depending on the reduced computational capacity of the embedded hardware devices (i.e., in practice, a microcontroller-based smart socket would not be able to perform as many requests per second as a desktop computer).

It is important to note that the aim of the tests was to assess empirically the framework performance limit for the above-mentioned hardware. Such an evaluation helped to analyze the behavior of the framework with increasing computational loads and made it possible to estimate the hardware that would be necessary for a real environment. To increase the capacity of system, it is possible to use more powerful hardware or scale the system horizontally. To do so, IoT devices can be divided into areas that are managed by cloudlets, which are closer than a cloud, so they manage less IoT devices and provide faster response times. However, such an architecture will require making use of specific protocols to coordinate the different cloudlets and thus sharing the information among them when it is required.

### 5.1. AR/MR Interaction Performance

In this first set of tests several requests (e.g., switch on/off commands) were sent to the smart power outlet through the REST API. Such requests are internally handled by the AR/MR-IoT framework by first sending them to Node-RED and then to the MQTT broker (Mosquitto). Thus, performance was measured from the moment the request was issued by the HoloLens to when it was parsed and executed by the smart socket.

The ’Interaction’ line in Figure 13 shows how response latency increases with a growing number of interaction requests per second. As it can be observed, interaction latency remains stable and under 200 ms for up to 50 requests per second. Then, for more than 50 requests per second, latency increases rapidly, reaching almost 6 s for 90 requests per second.

This behavior can be further analyzed by observing the ’Interaction’ line in Figure 14, which depicts the number of requests per second that the AR/MR-IoT framework is able to handle as the load of the server increases (for this second experiment, such a load was represented by an increasing number of simulated concurrent clients that performed interaction requests). As it can be observed in Figure 14, the number of interaction requests per second increases with the number of concurrent clients until it flattens between 200 and 300. For 300 concurrent clients the framework is able to process 746 interaction requests per second, but, after that point, the server becomes overloaded and starts to respond more and more slowly. This fact can be easily noticed from the server side, since its RAM memory usage increases rapidly.

Therefore, the previous experiments indicate that there is a throughput bottleneck on the framework: either Node-RED or Mosquitto slow down interaction request processing. To determine which of both is the responsible, it was measured Mosquitto throughput. The results are depicted in Figure 15 and show that the MQTT broker can handle more than 200,000 requests per second with no problem. As a consequence, it can be easily concluded that the bottleneck is Node-RED. In fact, the underlying problem is that, in contrast to Mosquitto, Node-RED was not specifically designed to handle a large number of concurrent requests.

The system throughput does not increase over 200 concurrent clients. This fact suggests that the system is not capable of supporting more concurrent requests. However, for more than 200 concurrent clients what it is relevant is the amount of requests per second that the framework is able to handle, since it establishes the limit of requests that different users can make per unit of time.

It is relevant to indicate that tests were finished at 500 test users because the throughput limit was already exceeded. Nonetheless, note that test clients emulate the requests made by a real client, but not the time intervals between each request (for these experiments, the emulated clients sent requests continuously in order to find the system performance limit). Therefore, the actual number of clients supported by framework will depend on the frequency with which such clients make requests.

### 5.2. IoT Data Update Performance

In a second set of tests, data collected from the sensors were requested using the REST API. It is important to note that in these experiments only the AR/MR-IoT framework REST API performance was measured when accessing the IoT data stored on the local database or in memory, so the MQTT broker was not involved.

Thorough tests were performed for a different number of concurrent clients. The ’Data request’ line in Figure 14 shows how many IoT data requests per second the framework is able to handle for up to 500 concurrent clients. Such a line shows that the system is able to handle up to roughly 1300 request per second from 150 concurrent clients and then becomes overloaded. In addition, it can be observed that the framework performance was clearly superior for this experiment than when processing interaction requests. This is due to the fact that IoT data update requests are answered directly by Node-RED, so it is not necessary to send a message over MQTT.

Finally, the ’Empty request’ line in Figure 14 represents the performance of the framework when it responds to IoT data update requests, but when such data are not collected from the local database, but from memory. Thus, the framework reaches 1500 requests per second for 150 concurrent clients, but then the framework becomes overloaded. Therefore, the obtained results show that the accesses to the database during IoT updates have a limited impact on the framework performance.

### 5.3. Practical Use Case Latency Analysis

There are two different types of response times relevant for the AR/MR-IoT application described in Section 4. The first one is related to the communication issued from the AR/MR application when sending a command to the smart power outlet. For sending such a command, the HoloLens glasses first send an HTTP request to the bridge service, which processes it and translates it into an MQTT command that is sent to the smart power outlet. The response time required for such a set of processes follows Equation (Equation 1), where:t_interaction_: it is the total interaction time required to send a command to the smart power outlet.t_HTTP_: it is the time required by the HoloLens glasses to perform an HTTP request that asks the IoT device to execute the sent command.t_MQTT_: it is the time required to send the command by using the MQTT protocol.t_bridge_ and t_smart_power_outlet_ are the times required by the bridge service and the smart power outlet, respectively, to process the command requests.

It is important to note that Equation (Equation 1) measures the interaction time of a one way request (i.e., the time that a user will perceive as the one required for executing the command on the smart power outlet (e.g., to switch it on or off)), so the request response time (i.e., the protocol acknowledgment of the execution of the command) is not taken into account for evaluating user experience.
(1)$$t_{\mathrm{interaction}}=t_{\mathrm{HTTP}}+t_{\mathrm{MQTT}}+t_{\mathrm{bridge}}+t_{\mathrm{smart}\_\mathrm{power}\_\mathrm{outlet}}$$

After measuring t_interaction_ for the proposed AR/MR-IoT application under normal operation conditions (i.e., when a HoloLens user performs less than a request per second), it was determined that the average latency when interacting with the smart power outlet was 96.16 ms, with a standard deviation of 17.24 ms.

The second type of response time that was measured on the proposed AR/MR-IoT application was related to data requests. For example, such requests can ask for the instant power consumption of a smart power outlet. In this type of requests, the bridge service cache comes into play by serving the information stored in its memory and by requesting periodically updated data from the IoT devices. This reduces considerably the latency as the system does not have to wait for the IoT device to answer the request since it is already cached.

The response times related to data update requests follow Equation (Equation 2), where t_data_update_ stands for the total data update latency, t_HTTP_ is the time related to the necessary HTTP exchange (it includes both the HTTP request and its response) and t_bridge_ is the processing time required by the bridge service.
(2)$$t_{\mathrm{data}\_\mathrm{update}}=t_{\mathrm{HTTP}\_\mathrm{request}}+t_{\mathrm{HTTP}\_\mathrm{response}}+t_{\mathrm{bridge}}$$

The average value obtained for t_data_update_ with the proposed AR/MR-IoT application under usual loads (i.e., for less than one request per second), was estimated in 95.64 ms, with a standard deviation of 38.85 ms.

As a conclusion, it can be stated that the obtained results show that the proposed AR/MR-IoT framework is able to achieve latencies lower than 100 ms when the server is not under a heavy load. This is enough to fulfill the requirements defined in Section 3.2 and also is within the reference values set by the ITU in [41], considering that the virtual elements shown on the screen of the AR/MR application are most of the time static (therefore, a latency below 10 ms is not required). It is also relevant to mention the work described in [53]: after a thorough analysis of the quality of the user experience for different video-games, the authors determine that such an experience can be considered as good for an interaction latency below 150 ms if the movement of the scenes is low or medium (as it occurs in most AR/MR applications).

Finally, it is worth mentioning that the proposed practical system was evaluated when considering a scenario where the server was hosted on a remote cloud. In case of needing faster response times, they can be further decreased by replacing the remote cloud by smaller cloudlets closer to the end devices.

## 6. Conclusions

This article presented an AR/MR-IoT framework that eases the integration of AR/MR and IoT devices. The framework makes use of well known open-source protocols and tools that together allow for communicating AR/MR and IoT devices dynamically and in real time. After reviewing the state of the art, this paper described the design and implementation of the framework, and provided thorough details on its use for a practical application: an AR/MR-controlled energy monitoring application based on a smart power outlet and Microsoft HoloLens. In order to evaluate the performance on the AR/MR-IoT framework, it was evaluated under different amounts of computational load. The obtained results show that, under normal operation conditions, the framework is able to respond in less than 100 ms to interaction and data update requests. However, for more than 50 requests per second, the operations performed by Node-RED suppose a latency bottleneck whose enhancement requires to manage data access faster.

## Figures and Tables

**Figure 1 sensors-20-03328-f001:**
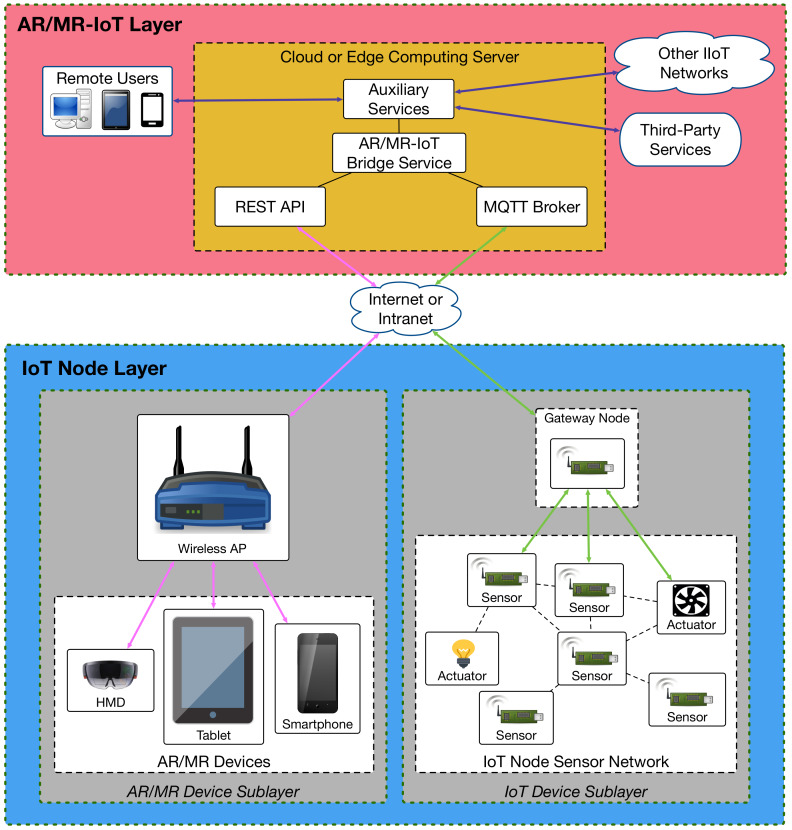
Communications architecture of the proposed system.

**Figure 2 sensors-20-03328-f002:**
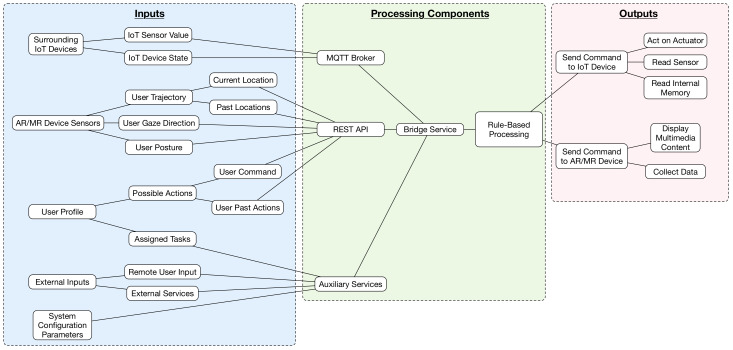
Component model of the proposed framework.

**Figure 3 sensors-20-03328-f003:**
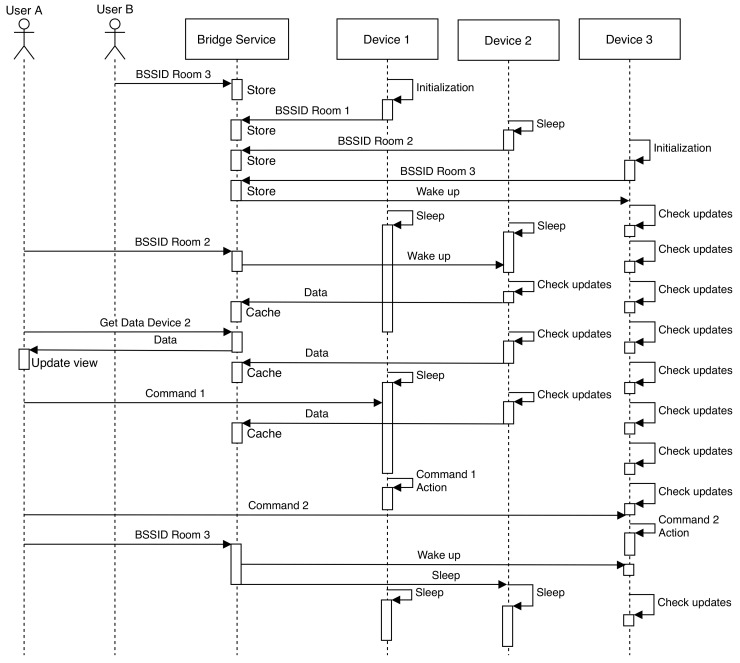
Sequence diagram of a low-power system built on top of the framework.

**Figure 4 sensors-20-03328-f004:**
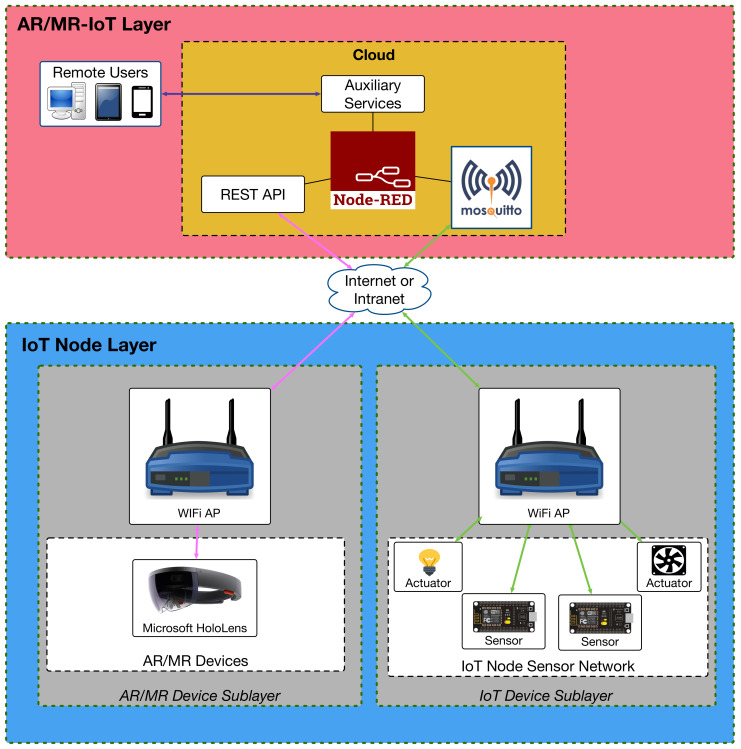
Implemented communications architecture.

**Figure 5 sensors-20-03328-f005:**
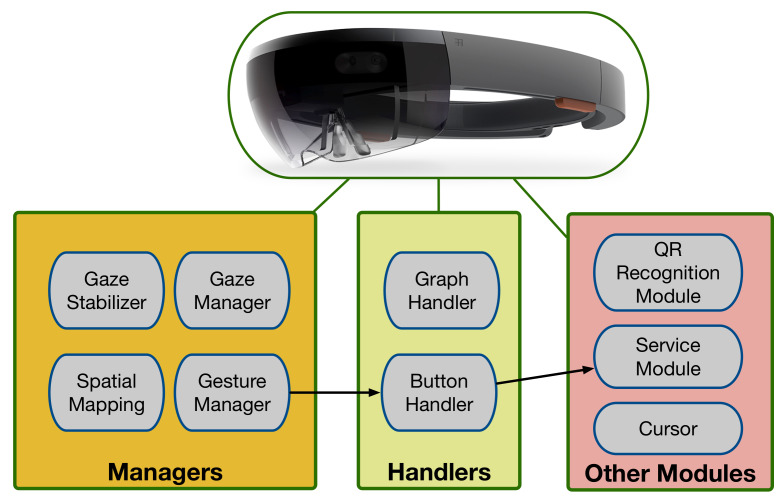
Main HoloLens modules used by the implemented application.

**Figure 6 sensors-20-03328-f006:**
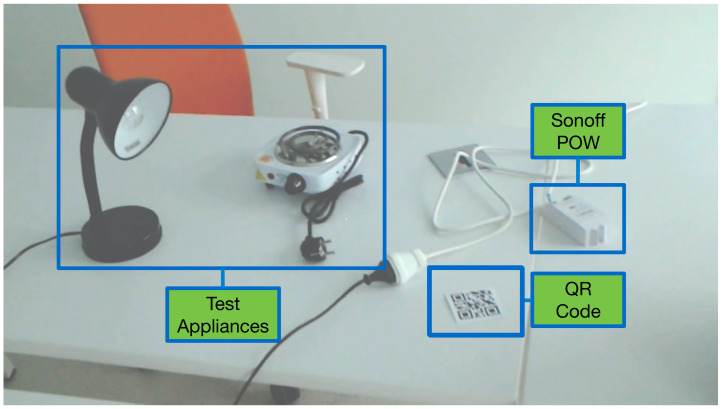
Smart socket setup.

**Figure 7 sensors-20-03328-f007:**
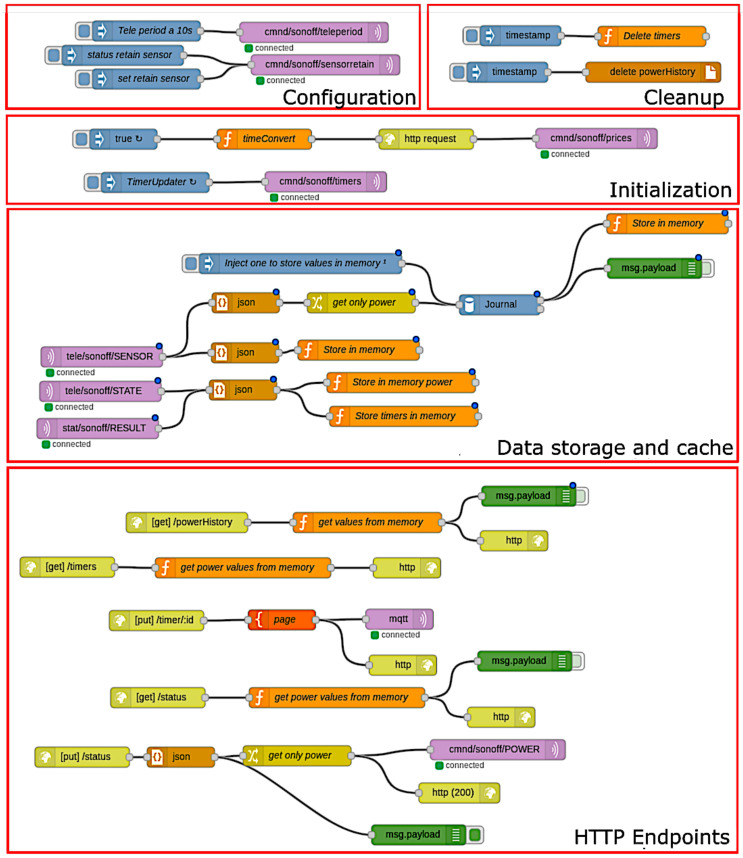
Screenshot of the data flows configured in Node-RED.

**Figure 8 sensors-20-03328-f008:**
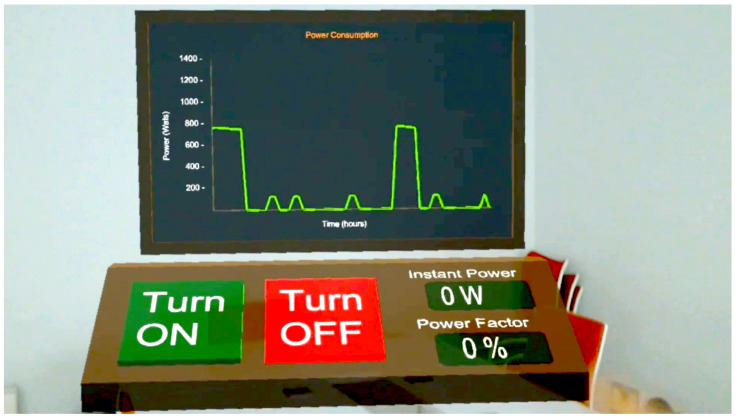
Virtual dashboard for controlling and monitoring the smart socket.

**Figure 9 sensors-20-03328-f009:**
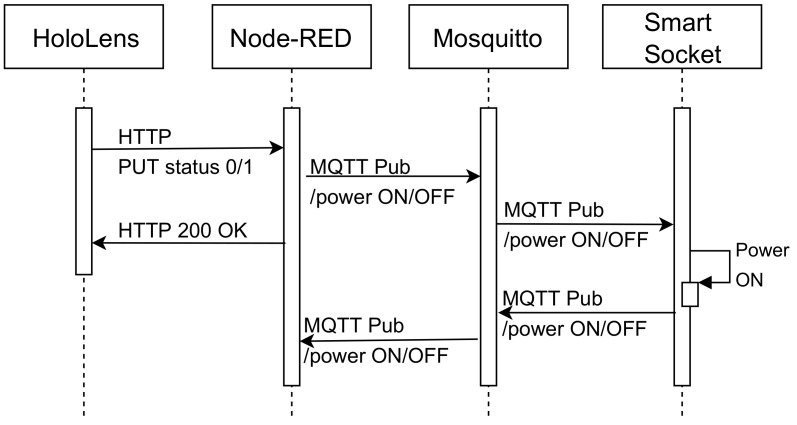
Sequence diagram for the switching ON/OFF of the smart socket.

**Figure 10 sensors-20-03328-f010:**
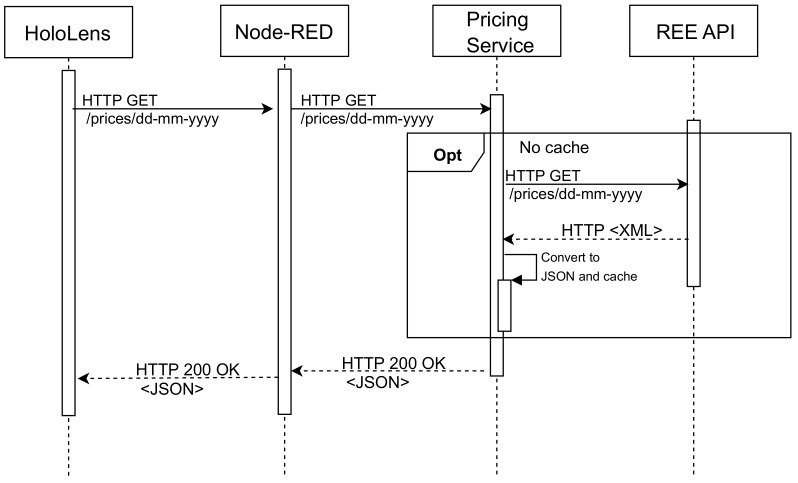
Sequence diagram for obtaining the energy cost per hour.

**Figure 11 sensors-20-03328-f011:**
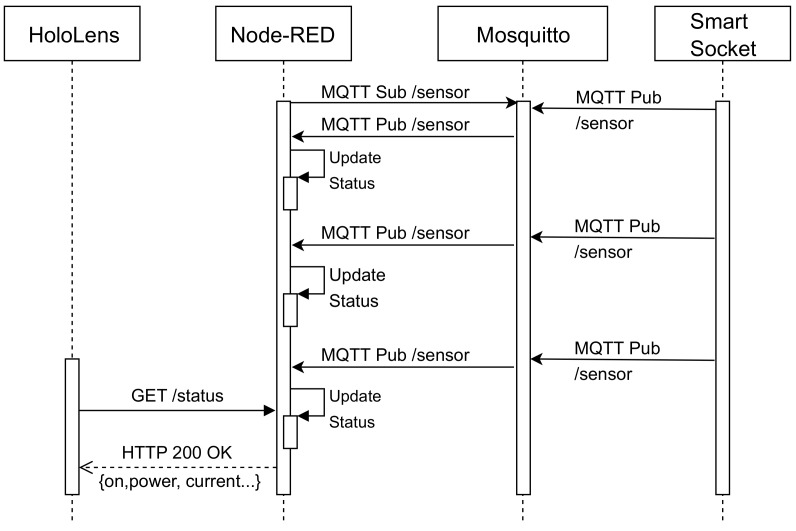
Sequence diagram for obtaining the status of the smart socket.

**Figure 12 sensors-20-03328-f012:**
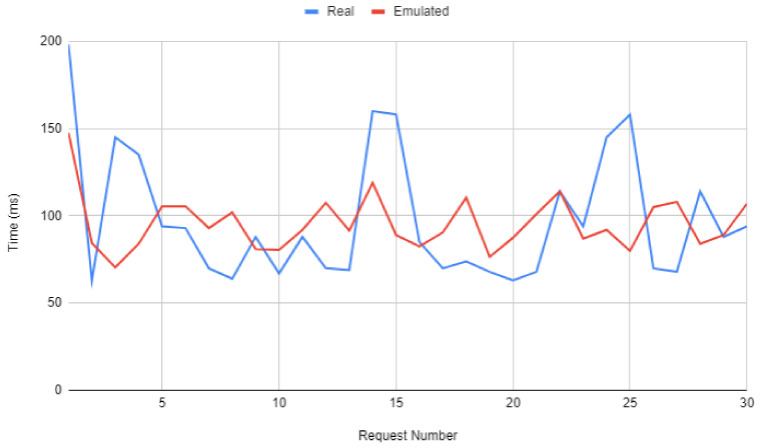
Response times for a real device vs an emulated device.

**Figure 13 sensors-20-03328-f013:**
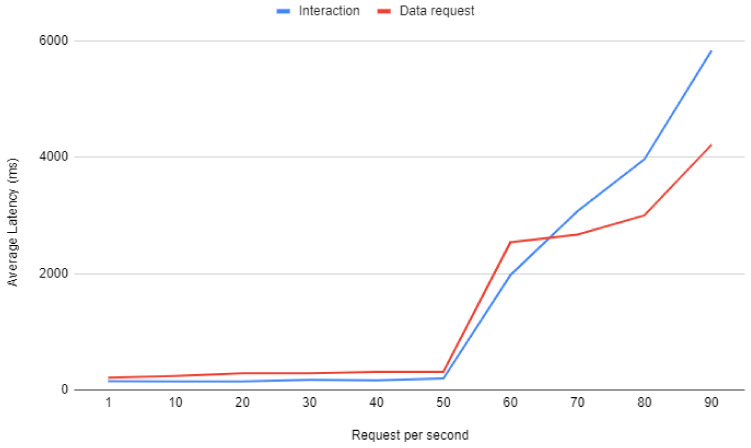
Response latency when increasing the load of the system.

**Figure 14 sensors-20-03328-f014:**
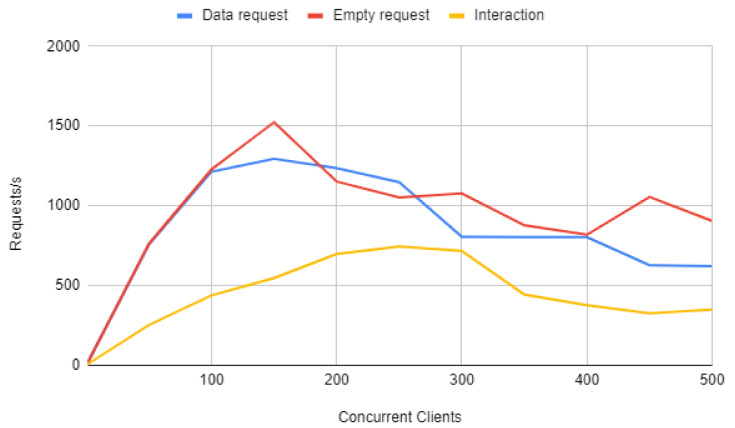
Throughput of the framework when increasing the load of the system.

**Figure 15 sensors-20-03328-f015:**
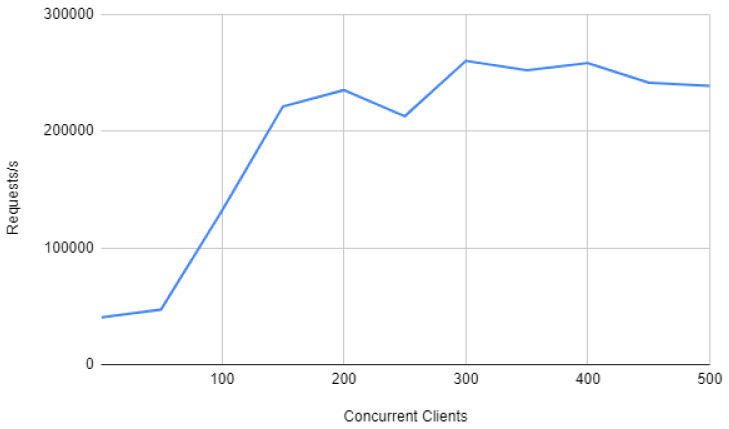
Mosquitto throughput.

**Table 1 sensors-20-03328-t001:** Comparison of the challenges addressed by the most relevant AR/MR-IoT frameworks.

Work	IoT Device Heterogeneity Support	AR/MR Device Heterogeneity Support	Open Source	Latency
Seungwoon Lee et al. [29]	Yes (OneM2M)	Yes (HTTP)	No	Not assessed
ARIoT [28]	Not specified	Not specified	No	Not assessed
VEoT [31]	Yes (HTTP)	Yes (HTTP)	No	Not assessed
Proposed Framework	Yes (MQTT)	Yes (HTTP)	Yes	<100 ms

**Table 2 sensors-20-03328-t002:** REST API endpoints.

Endpoint	Description	Exceptions
/powerHistory	Method: GET. It receives as a parameter a date in *dd-mm-yyyy* format and returns a JSON object with a list of the energy prices.	It returns ’404 Not Found’ in case there is no data for the requested day.
/timers	Method: GET. It returns a JSON object with the list of the 16 available *timers*.	If it is not configured, it returns the list of 16 timers as if they were inactive.
/timer	Method: PUT. It allows for modifying a timer configuration, so it receives as an input parameter the timer number.	
/Status	Methods: GET, PUT. It returns a JSON object with the status of the smart socket and allows for modifying it through a PUT request.	This method works in best-effort mode: the interface is updated before the node response is known. In case of error the state will be updated again as soon as possible to reflect a consistent state.

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
