# Peer review of "Creating the Internet of Augmented Things: An Open-Source Framework to Make IoT Devices and Augmented and Mixed Reality Systems Talk to Each Other†"

_sensors, 2020, doi:10.3390/s20113328_

Round 1

Reviewer 1 Report

There are some minor drawbacks in writing; the authors should carefully check the whole manuscript.

If possible, experiments should be conducted with real devices other than simulated ones.

For further research, the reviewer strongly suggest the authors to express the model in a formal way so that it can be logically modeled and inferred. Furthermore, the systematic evaluation method of user experience should be included in this framework.

Author Response

Dear Sir/Madam,

The authors would like to thank the reviewer for his/her valuable comments, which have certainly helped us to improve the manuscript. Please find attached our detailed responses to the comments. In order to ease the labor of the reviewers we have colored in red the differences with the previous version of the article.

Regards,
The authors.

Reviewer 2 Report

Presented solution has really low scientific level - especially with almost part theoretical known information. There are not described algoritmhs and methods, which can provide some research results. There is not implemented some special own developed communication standard or method for usage. There are not statictical results or verification of functionality of system. The developed practical system is just presented as assignment for illustration without research deep.

Author Response

(The authors gave the same response as above.)

Reviewer 3 Report

This paper presents a communication framework that allows AR/MR devices to send read/write commands to IoT devices over an HTTP server and a pub/sub network. It is trying to address a really interesting and challenging problem in the field. However, the paper doesn't specify any particular challenges this work is addressing, nor does its detailed design show that. (Trying to make all IoT devices speak the same protocol is not easy, and doesn't solve any fundamental research problems IMHO) The paper also does't make it clear what improvements this work brings compared to existing work, such as MQTT.

There are many challenges to bring interoperability to AR/MR and IoT. As an example, how to trade-off between responsiveness and energy consumption? Just using MQTT doesn't automatically solve this problem. pub/sub network helps saving energy for IoT devices by providing a DTN(Delay Tolerant Network)-like communication, as well as 1-to-many communication. But it doesn't provide a mechanism to tune this trade-off. A example contribution can be modifications on the platform to turn IoT devices to low-latency high-energy consumption mode when the users gets closer to them (by predicting where a user will look at, based on her/his location, attention, personalized preferences, etc.), and high-latency low-energy mode otherwise.

The evaluation should start with a practical goals of how scalable the system needs to be, instead of showing what the system happen to do with the particular implementation, operating system, hardware, and network environment. Some better questions to ask in evaluation are: (1) Does one server need to support 500 (or any particular number of) clients at the same time? (2) How to scale it if one server is not sufficient? (3) What does the trade-off between energy and latency look like on an IoT device? (4) what is a baseline latency and energy consumption, and why this work makes one of them, or both, better?

Author Response

(The authors gave the same response as above.)

Round 2

Reviewer 2 Report

Presented solution has low scientific level - theoretical known information is repaired. There are better then in version 1 described algoritmhs and methods, which can provide some research results. There is described own developed communication standard or method for usage. There are added statictical results or verification of functionality of system. The developed practical system is just presented as assignment for illustration with low research deep.

Author Response

(The authors gave the same response as above.)

Reviewer 3 Report

Thanks a lot for the detailed responses to my comments. I really appreciate the authors' efforts.

Unfortunately, I think the problems and challenges are still not articulated clearly enough.    I understand the problem is that AR/MR devices and IoT devices are not interconnected today, which is a well motivated research problem. However, the paper reads like it is trying to build a new standard and/or platform for this communication. It is unlikely others will adapt it without any questions, especially without addressing their needs and concerns in the paper.   To make this paper accepted, it must  1. clarify what challenges make it hard for AR/MR devices and IoT devices to talk to each other today. Is it because there are no needs? did people try but fail to make it work? The paper mentioned some related work, but doesn't mention why their approaches don't solve the problem. 2. articulate the design requirements, especially those can be measured and evaluated. For example, based on the paper, being able to support high-bandwidth and low-latency AR/MR data is required. Is there any other need? (It's ok if this is the only need and others failed to achieve it, but the paper must be clear about it) 3. discuss the design space and describe what choices are made based on the requirements. E.g., pub/sub is not sufficient to provide AR/MR communication, but is essential for IoT devices, so a mixture of both is required. 4. evaluate that the system meet the design requirements. It may not be an exact number of bandwidth or number of clients, but it must compare with others work or a reasonable baseline.

Author Response

Dear Sir/Madam,

The authors would like to thank the reviewer for his/her valuable comments, which have certainly helped us to improve the manuscript. Please find attached our detailed responses to the comments. In order to ease the labor of the reviewers we have colored in red the differences with the previous version of the article.

Best regards,

The authors.

Round 3

Reviewer 3 Report

Thanks for adding the challenges, design requirements, and evaluations on latency.